# The Barriers and Facilitators of Sport and Physical Activity Participation for Aboriginal Children in Rural New South Wales, Australia: A Photovoice Project

**DOI:** 10.3390/ijerph19041986

**Published:** 2022-02-10

**Authors:** Sarah Liew, Josephine Gwynn, Janice Smith, Natalie A. Johnson, Ronald Plotnikoff, Erica L. James, Nicole Turner

**Affiliations:** 1South Western Sydney Local Health District, Sydney 2145, Australia; sarah.liew@health.nsw.gov.au (S.L.); janice.smith1@health.nsw.gov.au (J.S.); 2Faculty of Medicine and Health, University of Sydney, Sydney 2006, Australia; 3College of Health, Medicine and Wellbeing, University of Newcastle Australia, Newcastle 2308, Australia; natalie.johnson@newcastle.edu.au; 4Hunter Medical Research Institute, Newcastle 2305, Australia; ron.plotnikoff@newcastle.edu.au (R.P.); erica.james@newcastle.edu.au (E.L.J.); 5College of Human and Social Futures, University of Newcastle Australia, Newcastle 2308, Australia; 6Rural Doctors Network, Sydney 2065, Australia; nturner@nswrdn.com.au

**Keywords:** physical activity, child, indigenous health, aboriginal, Australia, photovoice, yarning

## Abstract

Participating in physical activity is beneficial for health. Whilst Aboriginal children possess high levels of physical activity, this declines rapidly by early adolescence. Low physical activity participation is a behavioral risk factor for chronic disease, which is present at much higher rates in Australian Aboriginal communities compared to non-Aboriginal communities. Through photos and ‘yarning’, the Australian Aboriginal cultural form of conversation, this photovoice study explored the barriers and facilitators of sport and physical activity participation perceived by Aboriginal children (*n* = 17) in New South Wales rural communities in Australia for the first time and extended the limited research undertaken nationally. Seven key themes emerged from thematic analysis. Four themes described physical activity barriers, which largely exist at the community and interpersonal level of children’s social and cultural context: the physical environment, high costs related to sport and transport, and reliance on parents, along with individual risk factors such as unhealthy eating. Three themes identified physical activity facilitators that exist at the personal, interpersonal, and institutional level: enjoyment from being active, supportive social and family connections, and schools. Findings highlight the need for ongoing maintenance of community facilities to enable physical activity opportunities and ensure safety. Children held strong aspirations for improved and accessible facilities. The strength of friendships and the family unit should be utilized in co-designed and Aboriginal community-led campaigns.

## 1. Introduction

Australian Aboriginal and Torres Strait Islander people possess a rich and vibrant culture and have lived on and cared for the country for over 60,000 years [1]. The sudden disruption to lives and culture brought by British colonization in 1770 has created deep inequities and a high burden of poor health for Aboriginal and Torres Strait Islander people, which has been sustained until this day [2]. This inequity was sustained over the subsequent 200 or more years by a series of racist Australian policy eras resulting in marginalization, disadvantage, and extreme poverty [1]. One of the outcomes for Aboriginal and Torres Strait Islander people has been a decline in physical activity levels [3], contributing to poor health, including the development of chronic diseases such as type 2 diabetes [4]. Chronic diseases represent 70% of the gap in disease burden between Aboriginal and Torres Strait Islander people and non-Aboriginal Australians [5]. Over one-third of the total disease burden Aboriginal and Torres Strait Islander people experience could be prevented by modifying behavioral risk factors such as physical inactivity [6]. Here, we use the terminology ‘Aboriginal’ to refer to the Indigenous peoples of Australia (other than where Torres Strait Islander people are specifically mentioned in the references supporting this article), as this terminology is preferred by the communities participating in this study.

Whilst nationally Aboriginal children participate in more physical activity than their non-Aboriginal counterparts, this difference has been shown to decrease as children transition to adolescence [7]. Two studies conducted in New South Wales (NSW) reflect this activity decline [8,9]. Gwynn et al. reported that compared with their non-Aboriginal counterparts’ rural Aboriginal children aged 10–12 years were engaged in more physical activity [8]; however, by adolescence, physical activity participation rates were lower in a cohort aged 13–17 years (21% compared to 28%) [9]. A gender difference was also identified, with Aboriginal boys more likely to participate in physical activity than girls [9].

Aboriginal communities differ around Australia, not only by virtue of geographical location but also due to differences in factors such as language and culture [1]. It is therefore important to describe the experiences of Aboriginal children from different communities across the nation to gain insight into the breadth of experiences around participation in sport and physical activity and better inform relevant strategies and policies.

Five studies have reported Aboriginal young people’s perceptions about physical activity [10,11,12,13,14]. Of these, three (urban locations) explored children’s views of their physical activity in relation to type, amount, and the role this plays in their community [11,13,14]. Only two (rural and remote locations) explored physical activity barriers, neither in NSW [10,12]. The barriers identified in the latter studies included poor community facilities, lack of transport, costs associated with participating in physical activity, and experiences of racism [10,12]. Aboriginal adolescent girls were reported as feeling ‘shame’ ('stigma and embarrassment associated with gaining attention through certain behavior or actions' [15] (p. 8) and shyness wearing swimming costumes in pools and wearing sports clothes to exercise [10,12]. An established relationship between schools and the community was identified as a key facilitator to physical activity participation, as was the involvement and support of family and friends [10,11,12,13,14]. A recent study conducted with Torres Strait Islander communities found that community role models had a positive effect on some barriers to physical activity participation [16]. None of these studies were conducted in NSW, and given the cultural diversity between Aboriginal communities, it is yet to be established how applicable these findings are to young people in that state [17].

A recent systematic review of barriers and facilitators of sport and physical activity among Aboriginal and Torres Strait Islander children and adolescents found limited research (only nine studies) with a number of Australian states not represented [18]. The only study from NSW was not peer-reviewed and reported adult community members’ perceptions of the barriers and facilitators for children.

This study was conducted as a sub-study of the Many Rivers Diabetes Prevention Project (MRDPP) in response to that study’s findings regarding the physical activity of Aboriginal children [9,19]. The MRDPP aimed to improve the nutrition and physical activity of children living in the North Coast of rural NSW [19] and found physical activity among Aboriginal children declined over time with differences in patterns of decline existing between Aboriginal and non-Aboriginal children [9]. Despite tending to be more active in primary school [8], Aboriginal children from these communities recorded significant declines in non-organized, organized (winter only), and school activity over time when compared with their non-Aboriginal counterparts [9]. To gain insights into this finding and to inform future physical activity health promotion programs, the study team proposed exploring the Aboriginal children’s perceptions of barriers in their communities to sport and physical activity participation [19].

This study aimed to explore rural NSW Aboriginal children’s perceptions of the barriers and facilitators to their sport and physical activity participation.

The first author of this paper (S.L.) is a non-Aboriginal woman who completed an undergraduate (Honors) degree at the University of Sydney. J.G. is a researcher and non-Aboriginal woman who co-led the MRDPP with N.T. and has worked with the participating communities of this study for 17 years. N.T. is an Aboriginal woman from one of the participating communities who was the Manager Health Promotion and Senior Project Officer of the MRDPP. J.S. is an Aboriginal woman who is also from one of the participating communities and was an Aboriginal Project Officer of the MRDPP. R.P., E.J., and N.A.J. are researchers and non-Aboriginal co-authors who contributed their expertise in physical activity to this research.

## 2. Methods

### 2.1. Study Design

This study utilized a qualitative ‘photovoice’ methodology derived from the principles of participatory action research. The photovoice method requires participants to take photos which to them represent the topic or issue to be explored. Participants are then interviewed and asked to talk about the photos, typically discussing why these were taken and their meaning. The photos and interviews are the data used in the qualitative analysis. This method crosses cultural and linguistic barriers and enables participants to identify their community’s strengths and concerns [20]. Photovoice has been shown to be suitable and culturally appropriate for research with Aboriginal communities exploring issues as varied as food insecurity [21] and the experiences of Aboriginal health workers [22]. In this study, the photovoice method allowed children to explore the environmental and contextual factors that they perceived to influence their sport and physical activity participation [20].

### 2.2. Aboriginal Governance Structure and Ethics

The Aboriginal community governance structure and procedures that guided the MRDPP and this sub-study are described elsewhere [23]. Aboriginal Project Officers (APOs) employed in the MRDPP and from the participating communities led the design and implementation of this research, ensuring cultural safety [23]. The APOs also liaised with other organizations, contributed to the thematic analysis, and co-authored this publication. In writing this paper, authors applied the consolidated criteria for reporting qualitative research (COREQ) checklist. This was to ensure transparency with the research methods, and the important aspects of the process of this study were reported [24].

Ethical approval was received from the Hunter New England Local Health District Human Research Ethics Committee (reference number 11/10/19/4.04) and the Aboriginal Health and Medical Research Council of NSW (reference number 824/11).

### 2.3. Participants and Recruitment

Aboriginal boys and girls aged 10–14 years, residing in two communities (Community A and Community B) on the mid-north coast of NSW, were invited to participate. Recruitment was undertaken using a ‘snowball’ approach [25], with APOs contacting parents through the Aboriginal Corporation Medical Services (ACMS) in both communities. Parents were asked to inform their children of this study, and the children who were interested consented to participate. Consenting children then invited their peers to participate. Snowball sampling continued until no further potential participants could be identified [25]. Informed consent was obtained from all participants involved in the study.

A total of 26 Aboriginal children (12 girls and 14 boys) consented to take part in this study. Of these, 18 children attended the introductory session and were given cameras. A total of 17 children (9 girls and 8 boys) returned their cameras, and each participated in an individual yarn about their photos (Figure 1). The number of photos taken per child varied between 8 and 11. Thirteen yarns were audio-recorded, and hand notes were taken for the remaining four due to the community location of the yarn. In Aboriginal and Torres Strait Islander culture, a yarn is a relaxed and informal style of conversation that takes its own time, often flowing around a topic as information and stories are shared and then within the topic until the natural completion of the yarn [26].

### 2.4. Procedure

Parents of potential participants were handed recruitment packages with child and parent information statements and consent forms. Children who signed the consent forms were contacted by APOs via their parents and invited to attend an introductory group yarn in which the study aims, consent process, and study procedures were explained. Each child was provided with a digital camera and informed of its functions. Participants were given a week to take photos of the perceived barriers and facilitators to their physical activity participation in their community. The children also took photos of the physical activities that they enjoyed or wished to engage in. At the end of the week, yarning sessions were undertaken with each child in either a community location or the ACMS according to participant convenience and preference. These were conducted by APOs (J.S. and N.T.) or the lead investigator (J.G.), audio recorded or handwritten where the location was not conducive to audio recording, and audio recordings later transcribed for analysis. Children were invited to yarn about each of the photos they had taken, and these were uploaded to a secure location on the researcher’s computer. Prompts were co-designed with the APOs from the participating communities [27].

Once all individual yarns were completed, participants were then invited to a follow-up group yarn to select photos for community posters. Nine children and two parents (who were also aunties to other participants) took part in the first group yarn in Community A, and five children and one parent took part in the second (follow up) yarn to finalize their choices (Figure 2). ‘Aunty’ in Aboriginal culture is a term used to describe a respected female Elder in the community who may not necessarily be a family member [28]. In Community B, APOs reached consensus about which photos best reflected the themes arising from the individual yarns with children. Two children and two parents then met for a follow-up group yarn.

A repeated reflexive approach was taken throughout the process of finalizing photos deemed suitable for inclusion on posters. In Community A, photos were printed out by the research team and brought to the first follow-up group yarn. Children considered their photos and selected those that best represented their views of barriers and facilitators of physical activity. A parent or caregiver of each participant was present for this process. In Community B, due to local community factors at the time, children did not meet as a focus group to identify their selection. Here, the APOs considered the transcripts and handwritten notes, discussed each child’s photos, and reached consensus regarding those that best reflected the issues raised by the majority of participants in their interviews. Participants taking part in the group yarn concurred with the APOs reasoning and choice.

The final selection of photos (and related texts) was then considered for inclusion in several draft posters of differing designs by the research team. These posters were intended to be facilitators for community discussion of results. APOs invited all participants and their parents to take part in a poster design focus group in each community. Handwritten notes of the discussion were taken from these focus groups, which largely included parental feedback. To add richness to the findings, notes were cross-checked against key themes by the first author, and information relating to these themes was included.

### 2.5. Data Analysis 

Yarning transcripts and photographs were entered into a qualitative research software package NVIVO Version 11 (QSR International, Melbourne, Victoria, Australia) [29], for thematic analysis. Thematic analysis was informed by Braun and Clarke’s six stages, which involved data familiarization, initial coding and searching, and reviewing and defining themes [30]. To enhance the rigor of thematic analysis, S.L. and J.G. independently coded the first three yarns before discussing their similarities and differences. This aimed to reduce subjectivity that can occur when coding is completed by one researcher [31]. The remainder of yarns were coded by the first author. Codes were grouped together by looking at the relationships and connections between them to create categories and, subsequently, subthemes and overarching themes [30]. Preliminary themes along with the original transcripts and photos were sent to the APOs for their review and feedback (written and verbal). This feedback informed the final themes. Posters containing participants’ photos and final themes were co-created with the APOs.

### 2.6. Feedback of Study Outcomes to Communities

Results in the form of the posters and a verbal presentation with or without power-point slides were discussed at meetings with local city council representatives, key Aboriginal community members involved in the MRDPP, and members of the MRDPP Steering committee. Stakeholders were provided with a copy of the final MRDPP report to contextualize the conduct of this study [19]. Results were also presented for discussion at meetings of the Aboriginal Educational Consultative Groups (AECG) in both communities. Minor changes to wording in one poster were suggested and incorporated.

### 2.7. Socio-Ecological Framework

Physical activity participation is a complex behavior and is determined not only by the individual or their local environment but by ‘broader socioeconomic, political and cultural contexts’ [32] (p. ii10). A socio-ecological framework was applied to the barriers and facilitators identified by children to assist in understanding the scope of these complex factors and the ‘levels’ at which these exist in the participants’ environment. We applied the framework used in a recent mixed-methods systematic review of the barriers and facilitators to Aboriginal and Torres Strait Islander children’s participation in sport and physical activity [18] and coded the findings according to the levels they described: individual, interpersonal, community, and policy/institutional. In doing so, we aimed to align our findings and contribute to building evidence for practice.

## 3. Results

Thematic analysis revealed seven key themes (Table 1). Interviews and photos depicted a wide range of sports and physical activities enjoyed by the participants, including different types of football, bike-riding, basketball, soccer, running, and swimming. Photos largely reflected the barriers that participants experienced when accessing physical activity opportunities. 

### 3.1. Barriers

The physical environment was a key barrier to physical activity, particularly for Community A’s participants. Participants cited the littered and vandalized community facilities as a deterrent. Poorly maintained and run-down sporting venues were also reported, with tennis and basketball courts overgrown with grass and no usable equipment (Figure 3 and Figure 4). The poor state of these facilities prevented children from playing there despite their desire to.


*An ‘this is a photo of the basketball court. People used to drink there a lot and they used to like throw beer bottles and now it’s all wrecked because of them an’ the basketball nets are like, poles are like, falling, tilting, like it’s about to fall…*

*(P6A.)*


Participants discussed their experience of a lack of safety when engaging in physical activity due to hazards in the surrounding physical environment. The presence of litter such as glass in local playgrounds was identified by children as 'dangerous'. During the follow-up yarns, most children described continuing to play in playgrounds and parks despite it being unsafe.


*…and you can’t really see if there’s any glass or anything, so you never know when walking around in there. So, it’s not very safe.*

*(P2A.)*


The lack of designated space for children to engage in sports was identified by participants who also described playing non-organized sports in spaces such as near main roads. This supports children’s safety concerns around their physical environment and the lack of accessible and safe places to undertake physical activity.

Children identified consumption of unhealthy foods, including processed foods and sugary drinks, as a barrier to engaging in an active lifestyle. They discussed this factor as related to the development of obesity and diabetes, which, in turn, they perceived as having a negative impact on being active. Photos captured unhealthy foods on participants’ laps and signs of fast-food stores.


*[Soft drink] …it can stop us from playing games outside and it could give you diabetes and you can’t really like have what you want to eat sometimes... *

*(P7B.)*



*Well like junk food like would like stop you from a lot of sports, like putting on the weight and like things stuff like that. *

*(P9A.)*


The follow-up yarns expressed the view that the proximity and exposure of unhealthy food and drinks was a contributor to the consumption of these discretionary items. Children would pass the corner shop on the way to school, and high schools would sell sugar-sweetened beverages to students.

Participants acknowledged that engagement in excessive screen-based activities was sedentary behavior. In interviews, children acknowledged that screen-based activities displaced physical activity participation and recognized the impacts of this. Photos depicted different types of technology use, including iPads and computers.


*… sitting down …playing the play station or the phone instead of going out and being active… *

*(P5B.)*


The cost to participate and access physical activity opportunities was noted by participants. The high price of transport, sports registrations, equipment, and its maintenance were prohibitive for some parents. The cost barrier for parents hindered children from participating in their desired sport(s). In one photo (Figure 5), a participant held up a sign in front of a petrol station stating;


*Mum only has $5 left from her pay. I play at [a large regional city] that’s not going to get me there and back. *

*(P4B.)*


Handwritten notes from the second group yarns reported that parents were not aware of the funding and support that may be available to enable their children to participate in organized sport(s).

Lack of access to transport, both public and private, was associated with limited parental finances and availability of public transport, particularly when children lived out of town. Participants were reliant on parents or extended family members for transport to regular sporting competitions or community facilities. The availability of transport depended on family routine and dynamics. The issues with availability and affordability of transport were emphasized during the follow-up group yarns. Children discussed walking due to limited access to transport and this being the least-expensive option.

Five community-level, three interpersonal-level, and two individual-level barriers (Table 1) were identified when the socio-ecological model was applied. Children perceived barriers to participating in physical activity around: the physical environment, particularly the availability of safe and accessible community facilities; lack of parental finances to support sports participation; consumption of an unhealthy diet; and participation in sedentary activities.

### 3.2. Facilitators

Family members’ participation in sports and/or their sporting achievements were identified in both Community A and B as key factors facilitating physical activity, providing children with important role models for being active.


*…we started paddling out and I asked Dad if I could have a go. *

*(P9A.)*



*…my brother is surfin’ an’ we all love surfin’… *

*(P3A.)*


Family activities such as fishing were enjoyed on a regular basis.

Participants in Community A reported that school facilitated their engagement in regular physical activity. School events, such as the athletics carnival, encouraged children to engage in a variety of sports and to train for them in their own time. The provision of facilities such as the school oval gave children opportunities to engage in physical activity during lunch times.


*I don’t do any sports after school but um every lunch time I’m normally playing touch footy or I’m doing basketball, basketball with my friends. *

*(P1A.)*


Group yarns (Community A and B) reiterated these findings and discussed school as an important factor in helping children form an active lifestyle. The school was an environment that offered a wide range of opportunities to be active and an opportunity for children to engage in sport with their peers. Schools also enabled participation in physical activity through the provision of financial support and transport, both of which addressed factors described as barriers.

Participants enjoyed regular physical activity when they had access to adequate equipment and opportunities. In the final group yarns, participants were enthusiastic about outdoor play/non-organized physical activity as it was enjoyable, there was free choice of activities, and anyone could participate. Despite experiencing the complex barriers that made it difficult for children to be active, including gender role perceptions for one child, participants still desired to engage in physical activity.


*I took that picture like that cos it’s just saying that some kids actually wanna go in there and use it and stuff. *

*(P2A.)*



*Too old to play football because I am a girl, I still want to play football though. *

*(P3B)*


Participants proposed several suggestions to improve opportunities for physical activity in their community. This included better facilities and improved use of space by building community facilities.


*…the council should put ah real basketball court out the ridge cos we have a lot of space there. *

*(P3A.)*


Three interpersonal, and two each of individual, community, and institutional facilitators (Table 1) were identified when the socio-ecological model was applied. Facilitators were largely apparent at the individual and interpersonal level, with friends and family key facilitators. At the institutional level, schools were central to many children’s ability to take part in sports and physical activity. Children’s vision for improvements to their opportunities for physical activity was directed at the community level. They imagined facilities that better suited their community along with better use of space for community facilities.

## 4. Discussion

This study appears to be the first to explore rural NSW Aboriginal children’s perceptions of the barriers to and facilitators of their sports and physical activity participation. We found that the key facilitators of Aboriginal children’s physical activity exist at the interpersonal and institutional levels of the socio-ecological approach [18] and are physical activity engagement with friends, the strength of the family unit, and schools presenting opportunities for children to be active. The key barrier to physical activity participation identified by children was at the community level regarding poorly maintained community facilities and related safety issues. Other barriers perceived by participants included: intake of unhealthy foods, excessive screen time, inability to afford physical activity opportunities experienced as costly, and reliance on parents for transport.

The strength of the family unit as a key facilitator for physical activity aligns with the perceptions of Aboriginal children elsewhere [10,11,12,13,14]. Children discussed their family members (parents or siblings) who participated in sport and their sporting achievements as supporting and encouraging their physical activity. This factor is also a prominent facilitator for Aboriginal and Torres Strait Islander adults' physical activity participation [3]. Aboriginal people view physical activity as a collective occupation providing connections with others and the wider community [33]. Aboriginal families (parents and siblings) play a crucial role in supporting children and young people’s physical activity engagement through encouragement, role-modeling an active lifestyle, and facilitating activities involving exercise [12,13]. The lack of family involvement has been described as hindering children’s physical activity engagement in the Torres Strait and surrounding country [10].

Friends enable physical activity participation through the inherent enjoyment and fun experienced by children being active together in play, general activity, and sport [12]. Participants’ enjoyment and desire to participate in physical activity led them to hold aspirations for their community, including how space can be utilized to build community facilities such as a new basketball court. Enjoyment of sport and a desire to remain physically active have also been identified as facilitators to physical activity participation by Aboriginal adults [3,34]. As such, strategies to increase physical activity should explore options where children can also socialize with their peers or within an environment that encourages social connection.

School is experienced by Aboriginal children in this study as an environment that not only has better access to facilities and equipment but fosters socialization with friends. This aligns with findings elsewhere that have identified that an established relationship between schools and the community positively influences young Aboriginal people’s engagement in physical activity [12] and that Aboriginal children report school facilities and community events provide them with opportunities to be active [9,11,12].

Deteriorating community facilities and the resulting lack of safety reported by these NSW rural children expands on reports from studies in other Australian jurisdictions regarding rural Aboriginal children’s perceptions [12]. These factors present a significant deterrent to physical activity [35]. NSW state government policies and legislations control the availability and quality of community facilities and accessibility of neighborhoods, often through the actions of local councils that it funds [32]. Infrastructure in these communities is primarily funded by rates collected from residents [36]. As rates are calculated on property value [36], and the value of the property is less in the participating communities, fewer funds are available for infrastructure management. We suggest that the potential benefits of supplementing rates with additional funds be considered by local councils to ensure that infrastructure relevant for children’s health and wellbeing is adequately maintained in disadvantaged areas.

Participants in this study largely appeared to understand physical activity as engagement in organized sports, such as football, along with related non-organized sport/practice. The availability of relevant, accessible community facilities is therefore important. We note, however, that children did not consider the incidental exercise that takes place from day to day, such as walking to and from community facilities or walking as transport as physical activity. We call for local councils, communities, and schools to consider campaigns to promote alternatives to team sports, such as bike ridingand walking, to support children’s understanding that participating in such activities is also beneficial for their health. Such campaigns must be led by and co-designed with Aboriginal communities [27,37].

Children in this study identified the consumption of unhealthy foods and exposure to excessive screen-time as barriers to physical activity. Children described the association of these factors with low levels of physical activity and poor physical health, citing chronic diseases such as diabetes and obesity, both prevalent in their communities [2]. These have not been identified as barriers by young people in previous studies exploring Aboriginal and Torres Strait Islander children’s views on their physical activity [10,12] and should be harnessed in the design of future strategies to improve physical activity participation. Sedentary behavior due to time spent on screen-based activities is an issue for all children; however, a national report has found that Aboriginal children spend 25 min more on technology per day than their non-Aboriginal counterparts [7]. This is, therefore, a barrier that also warrants inclusion in programs that address children’s physical activity participation.

Participants described parental circumstances around vehicle availability and sufficient finance to afford car-associated costs as barriers to accessing sporting competitions or community facilities. This has also been identified by other young Aboriginal people as a barrier to accessing physical activity opportunities [12]. Transport disadvantage is common for Aboriginal people due to lack of access to and affordability of private and public transport options [38], particularly for those living in rural and remote parts of Australia. Lack of transport has been identified as a key barrier to physical activity and sports participation by Aboriginal and Torres Strait Islander adults [3]. The costs of public bus services in rural NSW have been found to be substantially higher than metropolitan areas and are more than residents are able to afford [39]. A lack of affordable and accessible transport places Aboriginal children at a further disadvantage when accessing physical activity opportunities. We suggest that local councils consider offering (or expanding) a community bus service to support weekend sport participation for children.

The inability to afford to participate in physical activity, including organized sports due to low income, has been noted by young rural Aboriginal people [12]. Aboriginal adults have also stated that the high cost of sports participation relative to their income is a very significant barrier to accessing physical activity opportunities [3,34]. While costs are also cited as a top barrier for other Australian children [40], additional financial barriers exist for Aboriginal people who experience socioeconomic disadvantage more than other Australians and possess a lower weekly household income compared to other households [5]. Associations between low physical activity levels and socioeconomic disadvantage have previously been identified [41], and the high costs associated with sport may contribute to low rates of physical activity for Aboriginal children and youth. In our study, parents indicated that they were not aware of local schemes through sports organizations or local councils to support the costs of children’s participation in sports. It has been suggested elsewhere that better promotion of sporting opportunities through local agencies and clubs to young Aboriginal people may influence physical activity participation [12].

The enduring impact of colonization on Aboriginal communities is an overarching driver of the barriers to physical activity participation identified in this study and was identified as such by the APOs on this study when discussing the results. The socioeconomic disadvantage and lower weekly income evident in many Aboriginal communities [5] have been acknowledged as enduring impacts of colonial government policies, which also included regulating income of Aboriginal people, forced disconnection from traditional land, forced removal of children, and marginalization of communities [42]. Marginalization included being required to live in settlements or missions ‘out of town’ and being either barred from entering a town or segregated if permitted to use facilities [1]. Poor community cohesion and racism were identified by Aboriginal parents from the participating communities as an ongoing barrier to their children being active [19], and also have their origins in colonial-government policies that disrupted and fractured communities [33].

Adopting the principles of co-design [27,36] when developing physical activity programs for Aboriginal children and ensuring that these programs are led and delivered by local Aboriginal community members [43] is recognized as imperative to improving the accessibility and cultural relevance of such strategies [23,33]. However, these approaches are still yet to be widely implemented:


*What has been missing from these… (government policies since 1989) … commitments is the genuine enactment of the knowledges that are held by Indigenous Australians relating to their cultural ways of being, knowing and doing. Privileging Indigenous knowledges, cultures and voices must be front and centre in developing, designing and implementing policies and programs. The sharing of power, provision of resources, culturally informed reflective policy making, and program design are critical elements *

*[44] (p. 1).*


Strengths of this study include the use of a novel method of investigating Aboriginal children’s perceptions of physical activity participation, allowing their voices to be heard. The participatory action research approach used in this research enabled a flexible response to participant and community needs and supported their engagement at all stages of the study. A reflexive approach to the final selection of photos allowed careful consideration of those that best represented participants’ views. The strong Aboriginal community governance structure enabled guidance on all aspects of the research process [23]. Community consultations allowed findings to be discussed with various Aboriginal community members who have been involved in the MRDPP and with local council representatives who wished for additional information. The posters distributed to community stakeholders allowed for further dissemination of results at a local level.

A limitation to this study was that a number of community-level events and challenges unrelated to the study emerged in Community B over the time that the yarns took place. These impacted recruitment numbers and children’s participation in follow-up yarns. However, the participation of APOs from the communities to some degree mitigated this issue, and feedback received from the community when results were presented was positive.

## 5. Conclusions

This photovoice study enabled Australian Aboriginal children from rural NSW to describe their experiences of sport and physical activity participation in their communities for the first time. Results extend the limited representation of Aboriginal children’s voices on this topic nationally. The identification of key facilitators at the interpersonal and institutional level and of barriers at the community level offer guidance for future strategies to address improvements in enabling Aboriginal children to participate more fully in the sports and physical activities that they aspire to. Prioritizing the maintenance of community facilities is important in enabling access to physical activity opportunities, and children held strong aspirations for improved and accessible facilities. Transport accessibility, along with the costs of sports participation, continue to be barriers to Aboriginal children’s engagement in sport and physical activity and require a whole-government response. The strengths of families and friendships should be harnessed to facilitate participation in sport and physical activity.

Barriers and facilitators identified by Aboriginal children are a result of the enduring impact of colonization on families and communities. Aboriginal community co-design and leadership of all matters of relevance to their communities, including in public health and health promotion, are essential and widely recognized as central to improvements in health and wellbeing [45]. However, the development of policies and programs that embody these approaches is only emerging, and implementation is yet to be fully understood and accepted. Only once this occurs will Australian Aboriginal children be enabled to wholly engage with and benefit from the sports and physical activity that they desire.

## Figures and Tables

**Figure 1 ijerph-19-01986-f001:**
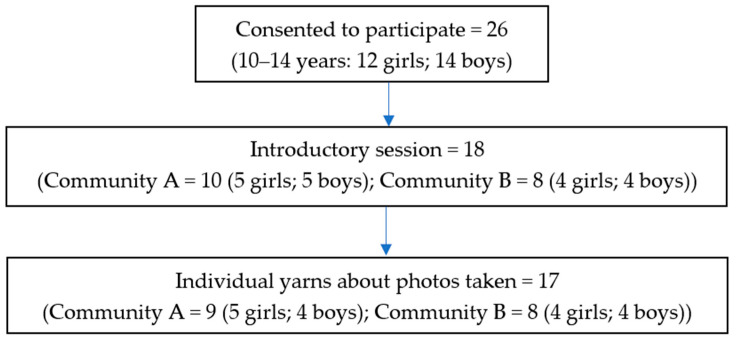
Participants.

**Figure 2 ijerph-19-01986-f002:**
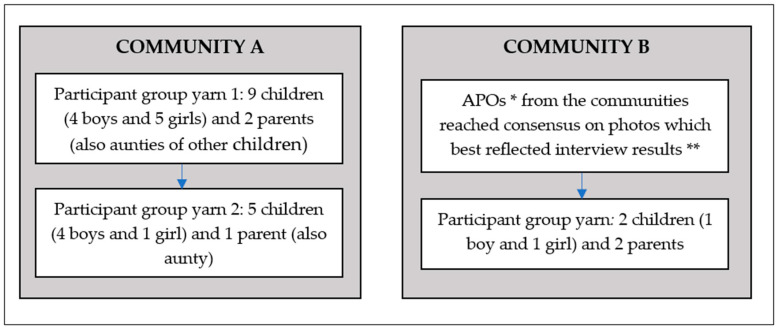
Process of photo selection for poster. Differed by community. * Aboriginal Project Officers; ** Due to community-level events and challenges, the first participant group yarn was unable to be conducted.

**Figure 3 ijerph-19-01986-f003:**
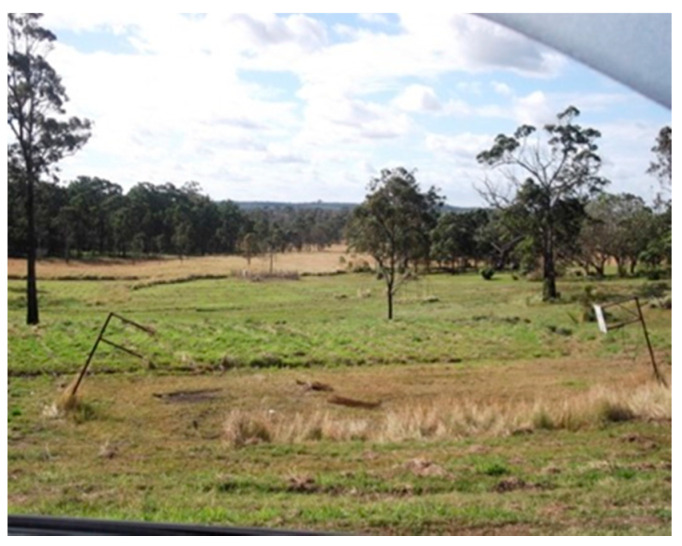
Community A’s basketball court.

**Figure 4 ijerph-19-01986-f004:**
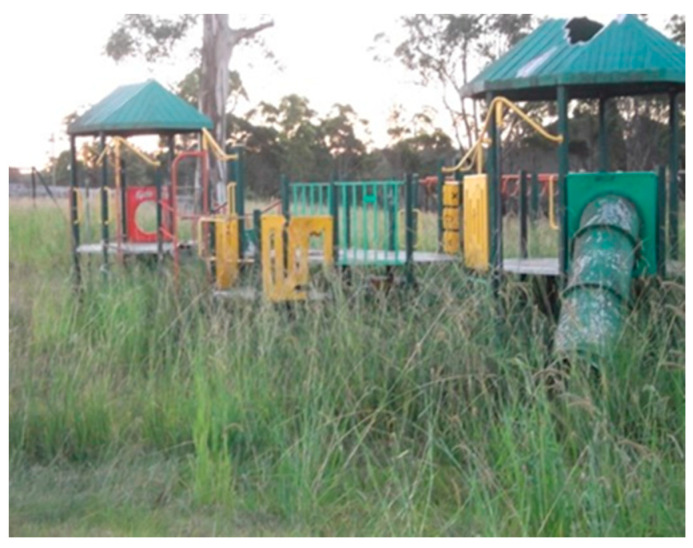
Community A’s playground.

**Figure 5 ijerph-19-01986-f005:**
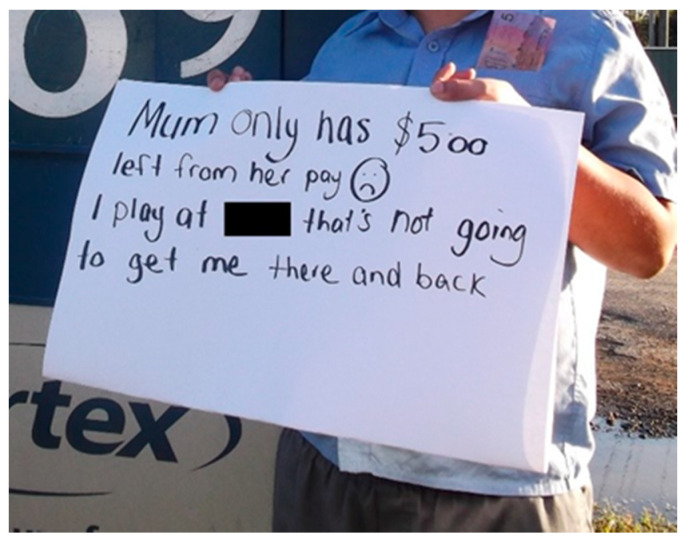
Associated transport costs.

**Table 1 ijerph-19-01986-t001:** Key themes of factors impacting physical activity participation.

Theme	Subtheme (Socio-Ecological Level [18])	Categories
**Barriers of Physical Activity Participation**		
*Physical environment*	Poor community facilities (C)	Poorly maintained Vandalized Littered Cracked footpaths Glass Frustration
	Lack of safety (C)	Arson incidents High volume of glass Deteriorating facilities Playing in non-designated areas
	Lack of designated space (C)	Bushland Playing near oncoming traffic/trains
*Individual risk factors*	Poor diet (I)	Unhealthy foods Impact on physical health Effect on ability to be active
	Excessive screen time (I)	Sedentary behavior Less time spent active Physical health impacts
*High costs*	Participation (C)	Sports registration and fees Equipment and maintenance Uniform
	Access to sports facilities (C, IP)	Lack of public and private transport Petrol
	Affordability (IP)	Lack of finances
*Reliance on parents*	Transport (C, IP)	Family routine and dynamics Unavailable to provide Inability to afford
**Facilitators of physical activity participation**		
*Social connections*	Family (IP)	Involvement Influence Role models
	Friends (IP)	Play sports at school Sense of community
*Schools*	Sporting events (In)	Exposure to different sports Opportunity to compete
	Opportunities provided (In)	Adequate facilities and equipment Lunch times Designated opportunities
*Enjoyment from being active*	Fun (I, IP)	Daily physical activity participation Others involved
	Desire (I, C)	Try different sports Use community facilities
	Visions for improvement (C)	Facilities to better suit community Better use of space for community facilities

Socio-ecological levels: I = individual; IP = interpersonal; C = community; In = Institutional.

## Data Availability

Restrictions apply to the availability of these data. Data was obtained from the participating Aboriginal communities and are available from the authors with the permission of the representatives of these communities.

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
