# Peer review of "The Barriers and Facilitators of Sport and Physical Activity Participation for Aboriginal Children in Rural New South Wales, Australia: A Photovoice Project"

_ijerph, 2022, doi:10.3390/ijerph19041986_

Round 1

Reviewer 1 Report

This is a well-written and engaging paper on an important and under-researched topic. The research was explained well, but I struggled with the ordering of some of the content (detailed below). Ethical and cultural considerations were respectful and well-considered. The conclusions drawn from the findings were well reasoned and the addition of policy implications was one of the strengths of the article.

Comments on the content

Methods section – more detail about what ‘photovoice methodology’ would be helpful for readers unfamiliar with this method.

I found the methods/results sections a little confusing – they were clearly written with a good level of detail, but some content did not seem to applicable to the section it was included in. Specifically, I thought the first few paragraphs of the Results section (including Figure 1) were more suited for section 2.3 ‘Participants and recruitment’ or 2.4 ‘Procedure’ in the Methods Section rather than in the Results section. Then, lines 232-240 and Figure 2 seemed to be about procedures, more suitable to include in section 2.4, not the Results section. I thought starting the results section off at line 244 would be clearer.

It was also not clear what role the community consultations played. Consulting’ implies more than information dissemination. What was the community being consulted about and if their views/advice was being sought, how were they ascertained and used within the project? It would therefore be helpful to clarify the purpose of the Community consultation.

I was also confused with the second paragraph of section 2.6 – this did not seem to be part of the Community consultations, but rather relating to coding/analysis which would be more suitable for sections 2.4 or 2.5? Or was the coding based on feedback from the community consultations?

There were some minor edits and style/formatting issues (some of which may be type-setting issues)

  1. I query the need for the inclusion of author initials in in-text references (e.g., at lines 39, 53, 67, 70 etc.). Usually these are only necessary when there are two different authors (or groups of authors) with the same surnames in the same year, but looking at the reference list this does not appear to be the case.
  2. Line 56 - Is a comma needed before and after however?
  3. Line 85 – suggest having NSW in full with (NSW) after for the first mention e.g., New South Wales (NSW); also in the abstract only the abbreviation is used. An international reader may not know what NSW is even though it is in the title
  4. Line 92 – apostrophe needed after members
  5. Line 150 – usually data saturation is reached not participant saturation; but since the group met together at the Introductory Session and collected data at the same time(?) determining data saturation within the research design would not be possible. Or were Community A and B run consecutively and therefore reaching data saturation was possible? Or, if it was participant crieria being met, what were these criteria to determine when recruitment was complete?
  6. Line 191 – space needed after NVivo 11 and (Stronach …)
  7. Line 214 – errors in page number for quote from Ball et al.?
  8. Line 217 – apostrophe needed after participants
  9. Inconsistency in hyphenation of follow-up - no hyphenation at lines 172, 511 but hyphenated at lines 175, 285, 264, 312
  10. Table 1 – ‘Playing near oncoming traffic/trains’ seems to fit with the “Lack of Safety” sub-theme and ‘Playing in non-designated areas’ seems more related the sub-theme “Lack of designated space’. Are these correctly placed in the table?
  11. Points 4 (Barriers) and 5 (Facilitators) are still results and should be a subheading of 3 (Results)?
  12. In the data anlysis section (2.5) and at Lines 290 – the term ‘interviews’ is used not yarns
  13. References – 3 and 4 (Allen et al.?) are these the same?; similarly for 42 and 43 (Verbiest et al) – delete duplicates
  14. Reference 30 (Macniven et al.) – Journal article should be capitalised; same for refs 42 and 43

Formatting/type setting issues

  1. Table 1 – the bullet points are very hard to read with the distracting indents -the table would be much more readable if they were all left justified
  2. Participants quotes are not obvious or discrete enough and just read like a new paragraph which is confusing. Similarly, the quote on line 491-497 is not clearly a quote
  3. References, section - titles of articles, journals etc are not italicised, nor is the journal volume, which according to the IJERPH style guide they should be?

Reviewer 2 Report

Dear authors,

The study presented by you is beneficial for practical use in sports. However, for improvement, consider the notes below.

Introduction

Row 39 – Citation (B. Allen et al., 2021), correct - (Allen et al., 2021a; Allen et al., 2021b)

Row 53, 100, 105, 131, 209 – Citation (J. Gwynn et al., 2010), correct - Gwynn et al., 2010

Row 54 – Citation - Gwynn and colleagues (2010), correct - Gwynn et al. (2010)

Row 76, 212, 435 – behaviour, correct – behavior

Methods

Row 137 – …that important aspects…, correct – those

Row 182 – The final selection of photos (and related texts) were then considered…, correct - was

Row – 188 – …information relating to these themes were included., correct - was

Row – 194 – rigour, correct - rigor

Results

In table 1 it is not clear the scale of found outcomes (marked by a dot for each indicator).

Discussion

The authors found that the facility is the main factor of physical inactivity of 17 participants. It seems to me that this finding is obvious at first glance and does not need scientific evidence. The suggestions made to improve the distribution of funds between the institutions are not substantiated and are not related to the research intent (Regarding the Rows 414 – 417.  

Row 418 – …activity as primarily engagement…, correct – primary

Row 485 – Adopting the principals of…, correct – principles

Row 496 – …program design are critical …, correct – is

Row 506 – …members who has been involved…, correct – have

Row – 530 – …promotion strategies is essential…, correct – are

Row 532 – Regarding this paragraph, It is a causal relationship?

„Only once this occurs will Australian Aboriginal children be enabled to fully engage with and benefit from the sport and physical activity that they desire.“

References

Row – 556, It was not cited in the text

Australian Bureau of Statistics. (2019). 4715.0 - National Aboriginal and Torres Strait Islander Health Survey, 2018-19. Re-556 trieved from https://www.abs.gov.au/ausstats/[email protected]/Latestproducts/4715.0Main%20Features22018-19?opendocument&tab-557 name=Summary&prodno=4715.0&issue=2018-19&num=&view= 558

Reviewer 3 Report

This is a significant and important study using innovative methodology.  Health outcomes for Aboriginal Australians compared to non-Aboriginal Australians demonstrate ongoing postcolonial difficulties for people who have a culture rich in physical activity and other health promoting practices.

Aboriginal researchers are included in the team and use of culturally sensitive practices such as yarning have been employed.  Photo-voice is one of the leading qualitative methods for understanding children’s perspectives.  It was clever to use this with yarning.

I wondered if it would be helpful to have ‘yarning’ as a keyword or mentioned in the abstract.  This may help other researchers who are interested in the use of yarning.

The procedures for analysis were clear and appropriate.  The findings were also clear and presented a series of potentially modifiable factors that could enhance the physical activity of the participants.

There seemed to be some vagueness about the ages of the participants and I wondered if this may have been due to the way data were collected.  Typically there is a mean age provided, for example.  My main interest here is not so much in the ages of the participants, but whether they attended primary or high school.  It would be great if this could be included in the information on participants for each community, gender and for each phase of the project.  Primary school and high school tend to have different emphases on sports and there may also be more emphasis on independence, such as walking to school, for children in high school.  There are also numerous studies showing declines in physical activity from primary to high school, including some Australian studies (e.g. Marks, Jennifer, et al. "Changing from primary to secondary school highlights opportunities for school environment interventions aiming to increase physical activity and reduce sedentary behaviour: a longitudinal cohort study." International Journal of Behavioral Nutrition and Physical Activity 12.1 (2015): 1-10. ) – although to my knowledge there are no studies with Australian Aboriginal children.  I therefore believe some readers will be interested in knowing how many participants were from high school and how many from primary school.

The last point of the Discussion, i.e. “A limitation to this study was that issues in Community B……..” could be explained more clearly.  I didn’t understand the point being made.

Figures 1 & 2 were unclear.  It would help if sharper images could be produced.  Table 1 needs better alignment. For Table 1, it may be that use of left justification is all that is needed.

The citation system used in not correct for IJERPH – see https://www.mdpi.com/journal/ijerph/instructions

I enjoyed the clarity of the writing.  I imagine it was difficult to achieve with such a complex project.  The effort is appreciated.
